# Biopsy Ratio of Suspected to Confirmed Sarcoma Diagnosis

**DOI:** 10.3390/cancers14071632

**Published:** 2022-03-23

**Authors:** Nasian Mosku, Philip Heesen, Gabriela Studer, Beata Bode, Vito Spataro, Natalie D. Klass, Lars Kern, Mario F. Scaglioni, Bruno Fuchs

**Affiliations:** 1Faculty of Medicine, University of Lucerne, 6000 Lucerne, Switzerland; nmosku45@gmail.com (N.M.); gabriela.studer@luks.ch (G.S.); mario.scaglioni@luks.ch (M.F.S.); 2Swiss Sarcoma Network, 6000 Lucerne, Switzerland; philip.heesen@uzh.ch (P.H.); beata.bode@patho.ch (B.B.); vito.spataro@hin.ch (V.S.); nataliedesiree.klaas@ksgr.ch (N.D.K.); lars.kern@ksw.ch (L.K.)

**Keywords:** sarcoma, biopsy, suspicion, confirmation, ratio

## Abstract

**Simple Summary:**

Determining the biology of mesenchymal tumor, imaging alone is usually not enough, and the final diagnosis is established through tissue analysis If the indication to perform a biopsy is not established frequently enough, an undesired unplanned resection of a sarcoma may result, and conversely, a patient’s discomfort as well as costs may increase. In here, using a real-world data registry of quality, we included the absolute number of a consecutive series of patients, to determine the prevalence of biopsies and its related diagnosis, to establish a reference, which may allow for the definition of a quality indicator for the work-up within a multidisciplinary team.

**Abstract:**

The ratio of malignancy in suspicious soft tissue and bone neoplasms (RMST) has not been often addressed in the literature. However, this value is important to understand whether biopsies are performed too often, or not often enough, and may therefore serve as a quality indicator of work-up for a multidisciplinary team (MDT). A prerequisite for the RMST of an MDT is the assessment of absolute real-world data to avoid bias and to allow comparison among other MDTs. Analyzing 950 consecutive biopsies for sarcoma-suspected lesions over a 3.2-year period, 55% sarcomas were confirmed; 28% turned out to be benign mesenchymal tumors, and 17% non-mesenchymal tumors, respectively. Of these, 3.5% were metastases from other solid malignancies, 1.5% hematologic tumors and 13% sarcoma simulators, which most often were degenerative or inflammatory processes. The RMST for biopsied lipomatous lesions was 39%. The ratio of unplanned resections was 10% in this series. Reorganizing sarcoma work-up into integrating practice units (IPU) allows the assessment of real-world data with absolute values over the geography, thereby enabling the definition of quality indicators and addressing cost efficiency aspects of sarcoma care.

## 1. Introduction

Sarcomas are malignant tumors of mesenchymal origin [1]. They account for 1% of all human cancers [2]. Sarcomas have an incidence of between 1 and 5 per 1,000,000 people [1], and are therefore considered a rare disease. Their diagnosis usually requires a high level of suspicion from the beginning of the correct and efficient work-up [3,4,5]. Once the diagnosis is established, not all suspicious lesions turn out to be malignant. According to Rowbotham et al. [6], who assessed all referrals to the sarcoma service in the UK, out of 49 patients who underwent biopsy with the suspicion of sarcoma, only 17 patients (35%) resulted with a malignant diagnosis, of which 13 (27%) were primary soft tissue sarcomas, 4 (8%) were soft tissue metastases (breast cancer, squamous cell carcinoma, colon carcinoma) and 32 (65%) were benign lesions (lipoma n = 19, posttraumatic lesions n = 5, vascular malformations n = 5, fibrous lesions n = 2 and nerve sheath tumor n = 1). In another study, Buvarp-Dyrop et al. [7] assessed the routes to diagnoses for suspected sarcoma in Denmark. Out of 545 patients, 102 (18.7%) were diagnosed as sarcoma and 68 (12.5%) as other malignancies, of these the most frequent being: metastasis (n = 30), lymphoma (n = 23) and myeloma (n = 6). The remaining 375 (68.8%) were benign lesions, the most frequent being: lipoma (n = 60), reactive tissue changes (n = 46), and schwannoma/neurofibroma (n =23). In another study made in the Rhone-Alpes region in France (Lurkin et al. 2010) [8], to evaluate the concordance between initial diagnosis and central pathology review for sarcoma cases, out of 366 patients diagnosed with sarcoma over a period of 1 year, 199 (54%) had full concordance between primary diagnosis (first pathologist) and second opinion (expert center pathologist), 97 (27%) had partial concordance (identical diagnosis of conjunctive tumor, but different grade or subtype), and 70 (19%) had complete discordance (different histological type or invalidation of the diagnosis of sarcoma). Another study (Gassert, F.G et al.) [9] analyzed retrospectively the histopathologic findings of 1753 patients presenting with a soft tissue lesion ≤5 cm. They found that 22.4% of these lesions were malignant.

The information about the relation between clinical suspicion of malignancy and the definitive histopathologic confirmation of sarcoma is important, because not all suspected sarcoma will ultimately be confirmed as such, but nevertheless absorb logistical and manpower capacities within the sarcoma work-up, besides causing inconveniences to the patients [10,11,12]. As long as the absolute number of biopsies for sarcoma suspicion as well as RMST are not defined, it will be impossible to determine whether too many or too few biopsies within an MDT actually are performed, and what for [13,14]. This may be particularly important in order to possibly understand the often frustratingly high rate of unplanned resections and potentially redefine the process of work-up of patients with mesenchymal tumors.

For these reasons, we are specifically asking the following questions:(1).How many consecutive biopsies were performed with the suspicion of sarcoma over a 3.2-year period (January 2018–March 2021) in our network?(2).How often was the suspicion of sarcoma confirmed as sarcoma?(3).What types of non-sarcoma lesions, so-called sarcoma simulators, were diagnosed?

## 2. Materials and Methods

Reorganizing the MDT into an integrated practice unit over the geography (IPU) [15], the Swiss Sarcoma Network (www.swiss-sarcoma.net, accessed on 23 February 2022, has established a prospective, real-world shared data sarcoma registry of quality, including the full longitudinal care cycle of patients over time. This registry focuses on the longitudinal assessment of the quality indicators of sarcoma care, with the aim of exchanging transdisciplinary and transparently sarcoma therapy relevant absolute data and of defining quality scores for sarcoma treatment [16].

All prospectively collected data from 1 January 2018 to March 2021 were included in this study for analysis. The data are stored with Adjumed and analyzed with the Adjumed Analyze tool (Adjumed Services AG, Zurich, Switzerland; www.adjumed.ch, accessed on 10 March 2022), which can be used for basic/descriptive statistics (such as combinations of parameters, and the extraction of data), as well as R statistical program (version 4.1.0). In this study, we have used the descriptive summary statistics and the two-sample differences tests.

All consecutive patients who have undergone a biopsy of any type to work up a suspicious mesenchymal mass during the indicated time period, independent of its anatomic location, and presented to the multidisciplinary team meeting/sarcoma board, were included in this study (https://swiss-sarcoma.net/pdf/GCP_1_minimal_workup_requirements.pdf, accessed on 17 January 2022).

Core biopsy is considered standard in this series; ultrasound-guided biopsy was applied for soft tissue tumors, and CT-guided biopsy was usually used with bone tumors. Excisional biopsies were performed when the non-lipomatous lesions were small (>2 cm) and epifascially located. Incisional biopsies were only indicated in exceptions. The diagnostic yield of the biopsies was >93%; for this analysis, only the first diagnostic biopsy per patient was included. In a study done in 2019 [3] to evaluate the diagnostic yield of the core biopsy in soft tissue lesions of the extremities, Qi, D. et al. [3] found a diagnostic yield of 96%.

All tissue specimens were assessed by a sarcoma reference pathologist according to the World Health Organization (WHO) classification of mesenchymal tumors and defined as benign, intermediate and malignant mesenchymal tumors, hematologic malignancy, metastasis and sarcoma simulators [11].

## 3. Results

### 3.1. Biopsies of Mesenchymal Tumors Performed over a 3.2-Year Period

Overall, 950 biopsies with the suspicion of sarcoma were performed during the study period. The biopsy specimens were collected from 950 different patients during the abovementioned time period. The type of biopsy was categorized into ultrasound (musculosceletic and visceral lesions), fluoroscopic (bone lesions) or CT-guided (thoracic, abdominal, pelvic lesions) core biopsy, fine needle aspiration, incisional and excisional biopsies [17,18], each separated into with or without (so-called whoops surgeries) suspicion of sarcoma (Table 1). In our study, sarcoma was confirmed in 62% of excisional biopsies, 57% in core biopsies, 53% in incisional biopsies and 48% in fine needle aspirations. The total amount of unplanned resections, i.e., so-called “whoops” surgeries, was 10% (21 incisional and 75 excisional biopsies without sarcoma suspicion) (Table 2).

The study found also similar RMST according to anatomic body regions. This ratio was 63% in the head and neck region, 61% in the trunk, 49% in the lower extremity and 47% in the upper extremity (Table 3).

Table 1 describes the 3 groups that resulted from the 950 biopsies and the subgroups for each of them.

Table 2 describes the types of biopsies performed and their respective percentages.

Table 3 describes differences in the RMST according to anatomic body regions.

### 3.2. Types of Tumors Identified through Biopsy

Overall, 55% (n = 522) of tumors were confirmed as sarcomas and, consequently, 45% (n = 428) of all biopsies turned out not to be sarcoma. Of the latter, 28% (n = 259) were benign lesions, and 17% (n = 169) were sarcoma simulators. Hence, the RMST was 0.55. Specifically, the final diagnoses included malignant (358; 38%) and intermediate mesenchymal tumors (164; 17%), benign mesenchymal tumors (259; 28%), metastasis (34; 3.5%), hematologic tumors (14; 1.5%) and sarcoma simulators (121; 13%) (Figure 1, Figure 2 and Figure 3).

### 3.3. Types of Sarcoma Simulators

Among the diagnoses other than mesenchymal tumors, 20% (n = 34) were metastasis, 8% (n = 14) were hematologic cancers, and 72% (n = 121) were sarcoma simulators.

Sarcoma simulators included 22 different types of lesions, the most common being inflammatory processes of soft tissues (15; 9%), degenerative processes of bones (14; 8%), granulomatous processes of soft tissues (11; 7%), pathologic fractures (9; 5%), hyperplastic granulation tissue (9; 5%), rheumatoid knots (9; 5%), epidermoid cysts (8; 5%), bone infarct (8; 5%) and periprosthetic inflammations (6; 4%) (Figure 4).

## 4. Discussion

This study analyzed 950 consecutive biopsies which were performed for sarcoma suspicion over a 3.2-year period. Of these, 10% were unplanned “whoops” resections. Of all biopsies, 83% were mesenchymal tumors and only 55% ultimately proved to be sarcomas. Of the non-mesenchymal tumors, 4% were other malignancies and 13% were sarcoma simulators with a wide variety of pathologies. Overall, the RMST in this series was 0.55.

There is only sparse information in the literature to include the analysis of a consecutive series of biopsies for mesenchymal tumor suspicions, and there is no analysis of sarcoma simulators specifically reported [19]. The largest study on the analysis of biopsies included 545 patients from Denmark and reported a rate of confirmed sarcoma diagnosis of only 19% (RMST 0.19), which is in contrast to the 55% (RMST 0.55) in this series [7]. The other study made in the Rhone-Alpes region in France, with 366 patients, reported a rate of 54% concordance between primary diagnosis and expert center definitive diagnosis [8]. One reason for this discrepancy may be explained by the lack of accepted defined categories to analyze and compare biopsy results. In addition, RMST depends on a multitude of other parameters. Such discrepancy in establishing the sarcoma diagnosis, however, evidences an unmet need in the work-up of sarcoma patients [9]. It may be helpful to agree on a common definition as to how to analyze biopsies to allow the comparison among multidisciplinary teams (MDTs). We believe that separating mesenchymal tumors (to define intermediate and malignant versus benign lesions) from non-mesenchymal tumors may be helpful [3] (Table 1). The latter category includes hematologic tumors as well as metastases from carcinomas, leaving a group with so-called sarcoma simulators [4]. In our MDT, for example, all subfascially located lipomatous lesions are biopsied, resulting in 114 mdm2-negative tumors and 51 atypical lipomatous tumors in our series, rendering an RMST for lipomatous lesions of 0.39 (39%). A comparison of RMST among various MDTs will be important to define a minimal percentage of conducted biopsies of lipomatous lesions for quality purposes [20]. In this context, it can also be speculated that there may be a minimal amount of sarcoma diagnosis established per diagnostic unit overall. This may reveal, for a respective MDT, whether patients undergo biopsies too frequently, or in contrast, not frequently enough, which conversely is indicated by the number of unplanned (“whoops”) resections [5]. Assessing the RMST of an MDT may serve as a quality indicator and ultimately also help to lower the unsatisfactory rate of unplanned resections [21].

The rate of unplanned resections has remained unchanged over decades [16,22], and the introduction of MDT per se may not have influenced this number either. There is currently no obvious strategy to address this issue. According to Abellan, J.F. et al. (2009) [19], in the 1990s, between 19% and 53% of the new patients seen in sarcoma centers were referred after an inadequate initial excision or whoops procedure. Another study (Pretell-Mazzini, J.) [14] pointed out that unplanned excisions of sarcoma occur in up to 50% of all patients with soft-tissue sarcoma. According to Zaidi, M.Y. et al. [16], considering the rarity of soft tissue sarcoma (STS) on the one hand, and the prevalence of benign soft tissue masses on the other hand, up to 50% of patients with STS will undergo a non-oncologic, unplanned excision for a mass initially presumed to be benign.

In this current report, the rate of unplanned resections was 10%, which is lower compared to the reports in the literature, and also lower compared to the rate of our own series before our prospective real-world shared data registry on quality was introduced. Obviously, the lower number in this report is explained by the inclusion of the total number of all biopsies, i.e., including all benign lesions (extrapolated on malignant tumors only, the rate of whoops surgeries was similar to international observations of 20%). On the other hand, based on a value-based geography model (VBGM) of care [23], the MDT herein represents multiple institutions and is therefore responsible for an entire region and not only for a single institution, thereby reaching all frontline care providers over a large geographic area.

When a biopsy for a tumor of the connective tissues is considered, the intention is to either confirm or exclude a sarcoma [24]. To the best of the authors’ knowledge, there is no detailed analysis of lesions which turned out to be non-mesenchymal tumors, which in this series totaled 17%. The majority of these were so-called sarcoma simulators, defined as benign lesions mimicking sarcomas clinically or on imaging. They comprise 13% of all biopsies performed and include mainly degenerative and inflammatory diagnoses, as illustrated in Figure 1. Whereas it is important to avoid unnecessary biopsies (to lower costs and potential complications) [25], the risk to not detect all sarcoma diagnoses has to be kept as low as possible. Assessing the sarcoma simulators of other MDTs will be instrumental to define a minimal RMST threshold, as a quality indicator of an MDT. Such information is also valuable to establish integrated practice units (IPU), to include the supra-regional work-up of potential sarcoma patients in the context of VBGM. For example, in dedicating resources, the hospital management must understand that of all lesions being worked up, only half of all patients ultimately need sarcoma care. On the other hand, such supra-regional IPU work-up units are considered the entry gate, allowing the assessment of the absolute number of patients, which is the base for real-world data. The VBGM of care creates new incentives and is associated with a transformation of healthcare delivery, to create a novel ecosystem. A prerequisite for such transformation of healthcare delivery, however, is the assessment of absolute patient numbers and respective interventions [26]. We therefore speculate that such novel ecosystem of care delivery, together with measuring the outcome using RMST, may reduce the rate of unplanned whoops resections.

There are some limitations of this study to be considered. Although the total number of biopsies is sizable, it remains unclear whether a larger cohort of patients may affect the results differently. It has to be taken into account that such analysis may vary from institution to institution, and from one healthcare system to another of respective countries [27]. The numbers for one single MDT presented herein, therefore, cannot be directly compared with other MDTs unless they are consecutive and represent absolute or real-world data (RWD) values. In order to define the ratio of suspected/confirmed sarcoma as a quality indicator, several MDTs would need to assess their own ratio for comparison and define a standard. Based on the absolute numbers, our study revealed some variations in RMST, both with respect for type of biopsy performed and anatomic regions. However, because of the sample bias, a superiority of one method over the other cannot be concluded in order to choose a type of biopsy, and may therefore not alter current practice of using core biopsy as standard of care [28].

To the best of the authors’ knowledge, this is the first study which focuses specifically on the ratio between sarcoma suspicion and ultimate sarcoma confirmation, with detailed analysis of sarcoma simulators. Assessing the absolute number of patients as RWD helps to define RMST, which may serve as a potential quality indicator of sarcoma work-up [29]. It may reduce the “scotoma” for unplanned “whoops” resections, which is considered an indicator of underestimation of the true number of sarcoma patients. On the other hand, infrastructure and personnel need to be dedicated to establish supra-regional sarcoma work-up units/IPUs to address and include absolute real-world data, thereby creating a novel ecosystem with a refined work-up approach [30]. This will ultimately allow sharing the experience between sarcoma networks independent of the geography.

## 5. Conclusions

The ratio of sarcoma suspicion to sarcoma confirmation is an important concept in the management of sarcoma patients. It is a quality indicator of the sarcoma work-up and may be a predictor for the cost efficiency of the care given to these patients. This ratio can be applied to all sarcoma MDTs, to increase the detection of sarcoma without increasing the number of unnecessary biopsies.

## 6. Patents

The results of this work and the new quality indicators that it has established are possessions of the Swiss Sarcoma Network.

## Figures and Tables

**Figure 1 cancers-14-01632-f001:**
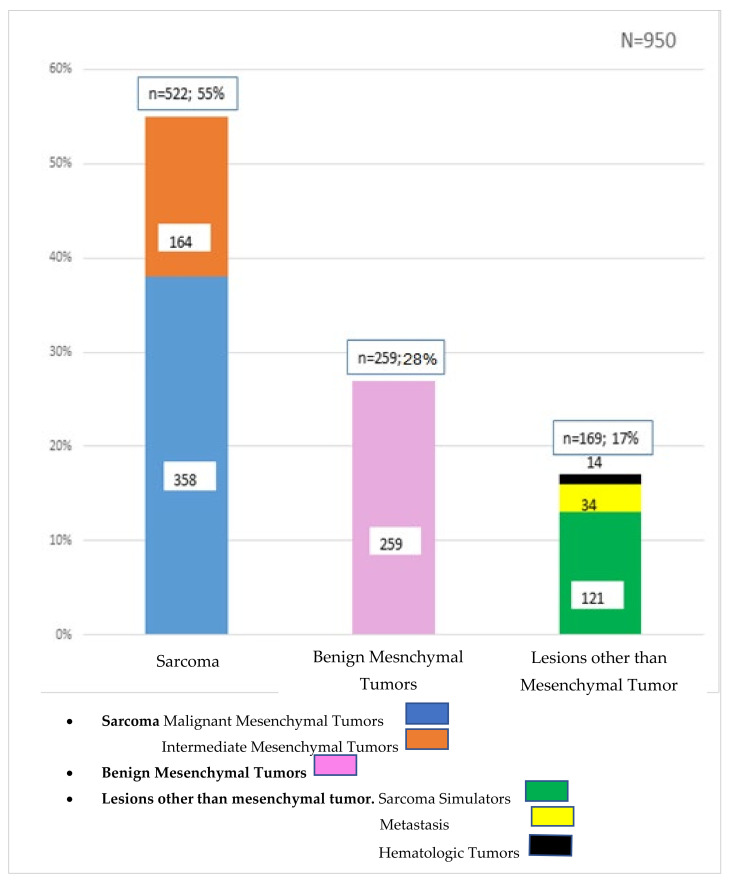
Types of lesions after biopsy.

**Figure 2 cancers-14-01632-f002:**
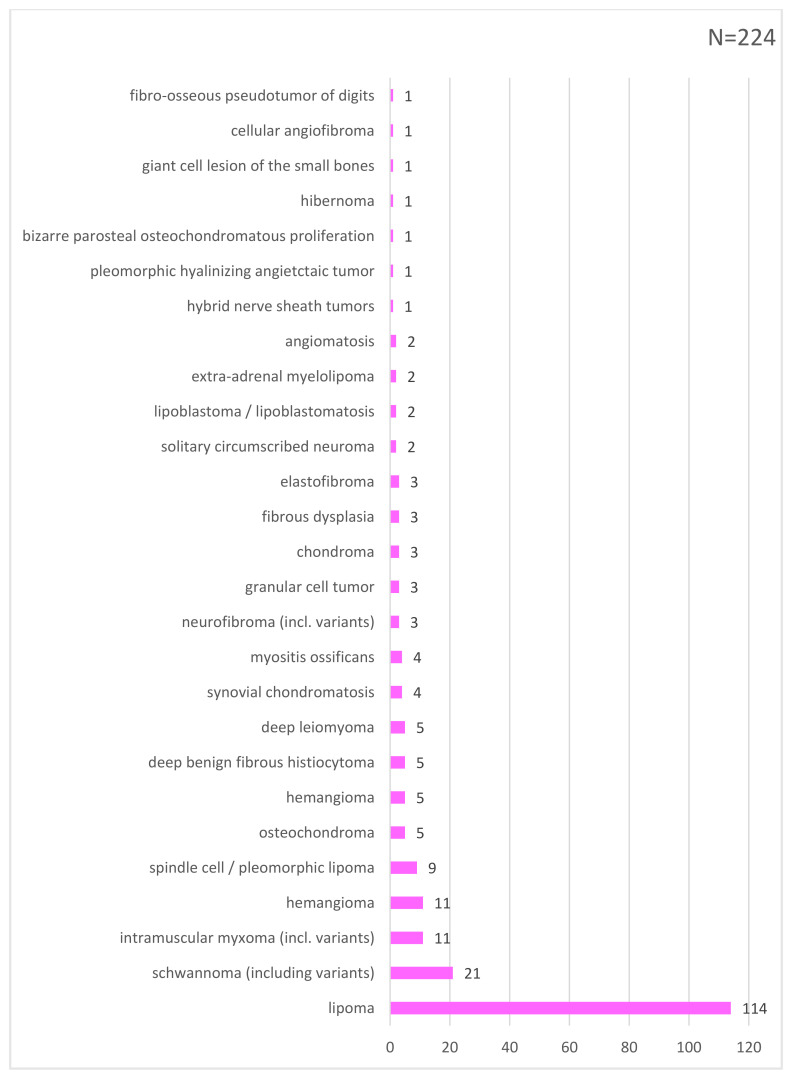
List of benign mesenchymal tumors.

**Figure 3 cancers-14-01632-f003:**
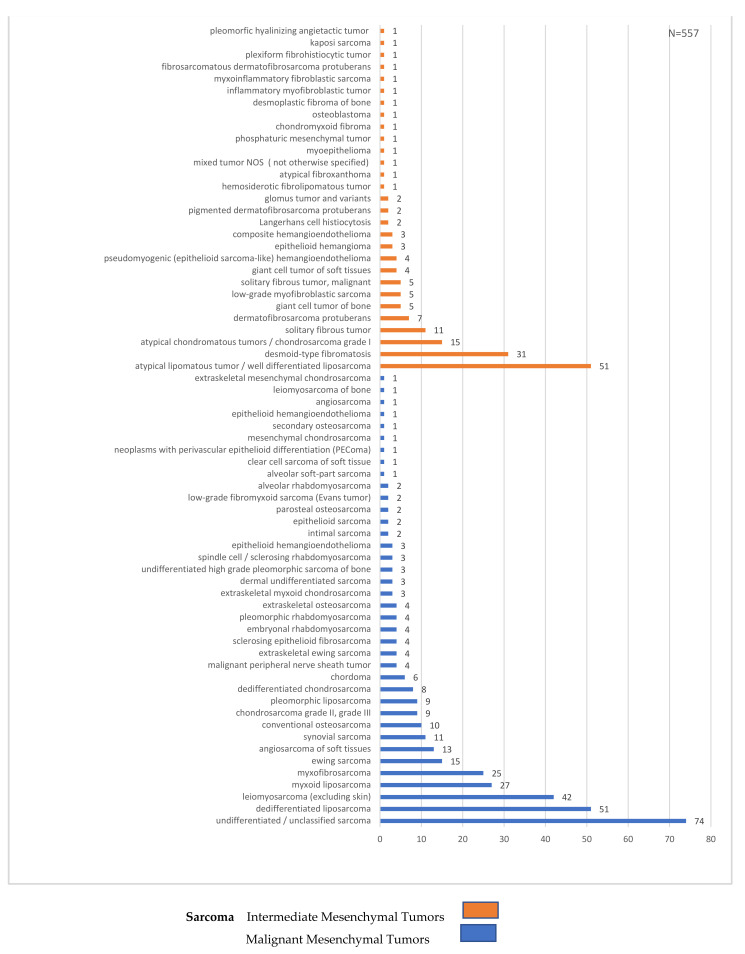
Histological types of sarcoma diagnoses.

**Figure 4 cancers-14-01632-f004:**
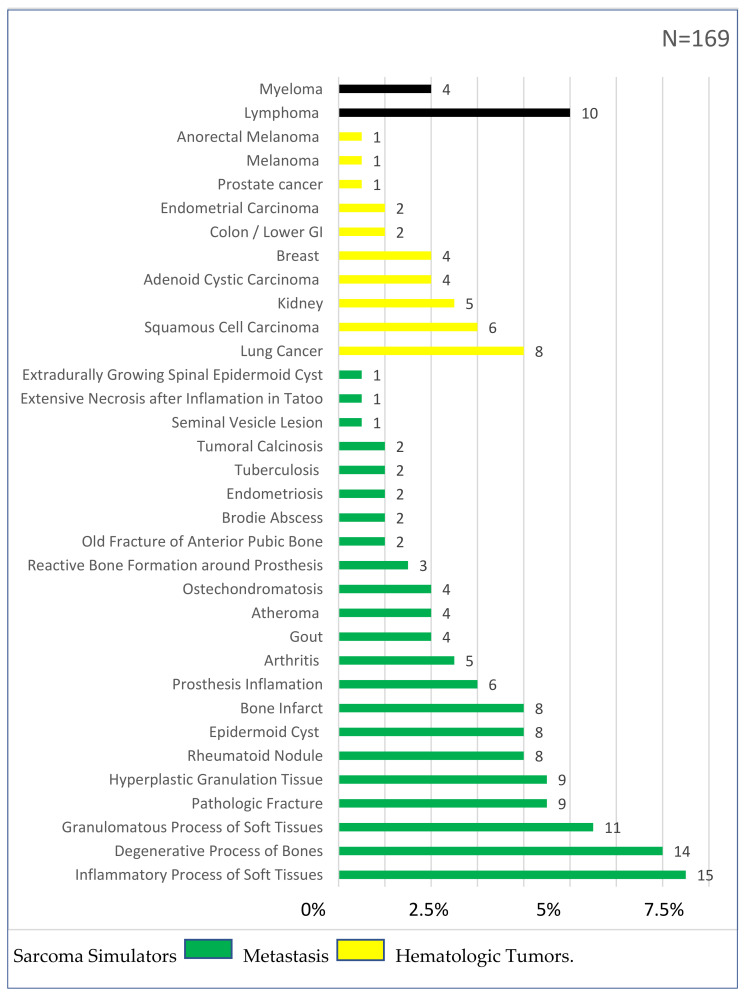
Types of lesions other than mesenchymal tumors.

**Table 1 cancers-14-01632-t001:** Classification of mesenchymal biopsies.

**Suspicion of Sarcoma**	Confirmed Sarcoma	Malignant
		Intermediate
	Benign Mesenchymal Tumors	
	Lesions other than Mesenchymal Tumor	Metastasis
		Hematologic Tumors
		Sarcoma Simulators

**Table 2 cancers-14-01632-t002:** Types of biopsies performed; n = 950.

Types of Biopsies		No./% of Cases	No./% of Confirmed Sarcoma
1	Core Biopsy	409/43%	542/57%
2	Fine Needle Aspiration	130/14%	456/48%
3	Incisional Biopsy with suspicion of sarcoma	110/11%	504/53%
4	Excisional Biopsy with suspicion of sarcoma	90/10%	589/62%
5	Incisional Biopsy without suspicion of sarcoma	136/14%	Not Applicable
6	Excisional Biopsy without suspicion of sarcoma	75/8%	Not Applicable

**Table 3 cancers-14-01632-t003:** RMST according to anatomic region.

Anatomic Regions Sarcoma Diagnosed	No./RMST
Head and Neck	598/63%
Trunk	579/61%
Upper Extremity	447/47%
Lower Extremity	466/49%

## Data Availability

https://www.swiss-sarcoma.net/ (accessed on 23 February 2022).

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
