# Peer review of "Biopsy Ratio of Suspected to Confirmed Sarcoma Diagnosis"

_cancers, 2022, doi:10.3390/cancers14071632_

Round 1

Reviewer 1 Report

In their manuscript, Nasian Mosku et al. reported the ratio of suspected to confirmed sarcoma diagnosis through biopsies. Due to the discrepancy in the clinical suspicion of malignancy and the definitive histopathologic confirmation of sarcoma, the authors sought to analyze the rate of malignancy in suspicious soft tissue and bone masses (RMST). They found that in 950 consecutive biopsies, 55% sarcomas are confirmed. Thus, this study might help increase the detection of sarcoma without increasing the number of unnecessary biopsies.

  1. The authors analyzed 950 biopsies in this study. Do these biopsies come from 950 patients, or one patient might have multiple biopsies included in the analysis? Please clarify this in the material.
  2. In Table 2, the authors concluded different types of biopsies included in this study. As one goal of this article is to understand the prevalence of biopsies and its related diagnosis, is there any difference in RMST between different biopsy methods?
  3. This study concludes multiple aspects of sarcoma diagnosis from biopsies. However, it’s hard to find a conclusion or a solution that could help to increase the diagnosis rate of sarcoma. Sarcomas can take place in different parts of body, such as arms, legs, trunk. Would RMST be different between biopsies taken from these parts?
  4. The writing of the article needs to be improved, especially the grammar and spelling.

Author Response

Thank you for your suggestions, we have taken each suggestion one by one, we have made the necessary changes in the document, and next to each change that we made we have put the suggestion of the reviewer which demands that change to be done. 

Reviewer 2 Report

For only one example, there is no detailed processing of the data regarding the size of the unclear soft tissue lesions and their correlation to the malignancy rate. Also a detailed processing of the data, such as the representation in the MRT imaging including the sequences used, is missing. The statement that no datas on this topic have been published in the literature in the past is incorrect. The publication by Wörtler et al. for example could be cited (BMC Cancer 2021). The analysis of the detailed entities is interesting in principle, but on the one hand there is no detailed processing and description, for example of the MRI characteristics, and on the other hand the conclusions drawn from this remain unclear. An added value through this work in the everyday treatment of sarcoma patients is not understandable and not reproducible from the drawn conclusions.

Author Response

(The authors gave the same response as above.)

Reviewer 3 Report

Dear Authors,

The research is original and important.

I appreciated the paper, I suggest You several improvements.

Majorn concerns:

  • Methods are really poorly described. Please, improve this section.
  • How biopsies have been performed? Usually US-guided for soft-tissue neoplasms, CT-guided (or fluoroscopic) for bone neoplasms, or surgical procedure biopsies. Implement the use of image-guidance, in discussion and methods section. Specify this. Core-needle is very vague.
  • It is not enough to explicit that you used 'R' program for statistics. Report the tests used.
  • Bibliographic section is really poorly. Implement references.
  • It would be nice to analyze (if you have data) and discuss (in regards to literature) about the diagnostic yield of these biopsies (there were non-diagnostics specimen?).

Minor comments:

  • Please substitute the terms 'mass'. Especially in case of skeletal lesion, it is not a mass a not. 'Neoplasm' should be the preferred term.
  • Please indicate the reference at the end of the sentences.

Author Response

(The authors gave the same response as above.)

Round 2

Reviewer 1 Report

The authors have addressed the reviewer’s comments well. The paper quality has greatly improved with more details and clarifications. I recommend the paper for publication in the present form.

Reviewer 3 Report

I am satisfied with the revisions performed, thank You!